# Ozone formation sensitivity based on the secondary formaldehyde-tonitrogen\_dioxide ratio (FNR<sub>sec</sub>) derived from ground-based remote sensing measurements and a chemical transport model

Nguyen Doan Thien Chi<sup>1,2</sup>, Hiroshi Tanimoto<sup>1,2</sup>, Satoshi Inomata<sup>2</sup>, Kohei Ikeda<sup>2</sup>, Yange Deng<sup>2</sup>, Royston Uning<sup>2</sup>, Tamaki Fujinawa<sup>2</sup>, Astrid Müller<sup>2</sup>, Masatomo Fujiwara<sup>3</sup>, Shungo Kato<sup>4</sup>, Hisahiro Takashima<sup>5</sup>

- <sup>1</sup> Graduate School of Environmental Studies, Nagoya University, Furo-cho, Chikusa-ku, Nagoya 464-8601, Japan
- <sup>2</sup> National Institute for Environmental Studies, 16-2 Onogawa, Tsukuba, Ibaraki 305-8506, Japan
- <sup>3</sup> Faculty of Environmental Earth Science, Hokkaido University, Sapporo 060-0810, Japan
- <sup>4</sup> Faculty of Urban Environmental Sciences, Tokyo Metropolitan University, 1-1 Minami-Osawa, Hachioji, Tokyo 192-0397,
   Japan
  - <sup>5</sup> Faculty of Science, Fukuoka University, 8-19-1 Nanakuma, Jonan-ku, Fukuoka 814-0180, Japan

Correspondence to: Nguyen Doan Thien Chi (nguyen.doanthienchi@nies.go.jp) and Hiroshi Tanimoto (tanimoto@nies.go.jp)

Abstract. Sensitivity analysis is essential for developing effective ozone (O<sub>3</sub>) mitigation strategies. This study aims to extensively investigate the diurnal, seasonal, and vertical chemical sensitivity of O<sub>3</sub> using a photochemical indicator, the secondary formaldehyde (HCHO)-to-nitrogen\_dioxide (NO<sub>2</sub>) ratio (FNR<sub>sec</sub>) as measured by Pandora remote-sensing spectrometers located across Japan. Region-specific FNR<sub>sec</sub> thresholds were determined using the GEOS-Chem chemical transport model. Surface concentrations and vertical column amounts of HCHO and NO<sub>2</sub> were obtained from in situ measurements and Pandora spectrometers. The concentrations of HCHO and NO<sub>2</sub> varied with time of day, season, and altitude. Moreover, external pollution transport affected the vertical profiles and likely contributed to elevated concentrations. Seasonally, the ozone sensitivity analysis showed that NO<sub>x</sub>-limited conditions were dominant in summer, transitional regimes in spring and fall, and VOC-limited regimes in winter. Vertically, VOC-limited conditions typically formed near the surface layers, followed by transitional regimes in the mid-levels, and NO<sub>x</sub>-limited regimes aloft. Therefore, O<sub>3</sub> mitigation strategies should target not only the surface level but also elevated altitudes. This study contributes to fostering a comprehensive understanding of O<sub>3</sub> sensitivity in the troposphere using FNR<sub>sec</sub> retrieved from Pandora measurements.

Keywords: ozone chemical regime, Pandora, FNR<sub>sec</sub>, GEOS-Chem model.

# 1. Introduction

Tropospheric ozone (O<sub>3</sub>) is a central secondary pollutant formed through photochemical reactions involving its main precursors: nitrogen oxides (NO<sub>x</sub>, including NO and NO<sub>2</sub>) and volatile organic compounds (VOCs). Increases in tropospheric O<sub>3</sub> levels negatively affect human health (Liu et al., 2018; Nuvolone et al., 2018), crop productivity (Mahmood et al., 2020; Ramya et al., 2023), and ecosystems (Feng et al., 2021; Grulke and Heath, 2020). Due to its well-known impacts, enormous efforts have been made in many cities and countries to mitigate O<sub>3</sub> pollution (Hu et al., 2024; Chang et al., 2025; Shi et al., 2025a). A major challenge that hinders O<sub>3</sub> pollution mitigation strategy is that O<sub>3</sub> formation results from non-linear photochemical reactions of its precursors, rather than from direct emissions (Mishra et al., 2024; Sadanaga et al., 2017). Additionally, long-range transport makes O<sub>3</sub> pollution even more difficult to control (Huang et al., 2010; Nguyen et al., 2022). Sensitivity analysis is of significant importance for developing effective O<sub>3</sub> mitigation strategies. Tropospheric O<sub>3</sub> production is conventionally categorized into three regimes: NO<sub>x</sub>-limited (or NO<sub>x</sub>-sensitive) regime, transitional regime, and VOC-limited (also referred to as radical-limited, VOC-sensitive, or NO<sub>x</sub>-saturated) regime. Depending on the sensitivity regime, controlling either NO<sub>x</sub> or VOC emissions can be an effective approach to mitigating O<sub>3</sub> pollution. Sensitivity analyses have been conducted using various approaches, including model-based methods (Thorp et al., 2021), absolute sensitivity analysis (Sakamoto et al., 2019), and photochemical indicators such as the formaldehyde-to-nitrogen dioxide ratio (FNR)

https://doi.org/10.5194/egusphere-2025-5266 Preprint. Discussion started: 14 November 2025 © Author(s) 2025. CC BY 4.0 License.

(Jung et al., 2022; Qian et al., 2024; Souri et al., 2023) and the non-methane-hydrocarbon (NMHC)-to-NO<sub>x</sub> ratio (Dhanya et al., 2024; Sharma et al., 2021). Among these, FNR is considered to be one of the most precise indicators for assessing chemical sensitivity. O<sub>3</sub> formation sensitivity can be visually represented using empirical kinetics modeling approach (EKMA) that produces curves between NO<sub>x</sub> and VOCs (Tonnesen and Dennis, 2000). However, measuring hundreds of VOC species is impractical. Meanwhile, HCHO reflects the VOC oxidation strength and is widely used as a proxy for VOCs (Irie et al., 2021). It should be noted that only secondary HCHO, produced photochemically from VOCs, accurately reflects the VOC oxidation capacity (Xue et al., 2022). Primary HCHO is directly emitted from anthropogenic activities; therefore, considering primary HCHO may be misleading in the assessment of O<sub>3</sub> chemical sensitivity. Previously, the HCHO-to-NO<sub>y</sub> ratio was used as an indicator of the chemical sensitivity. NO<sub>y</sub> consists of NO<sub>x</sub> and NO<sub>z</sub> (including HNO<sub>3</sub>, HONO, organic nitrates, etc.). More recently, the HCHO-to-NO<sub>2</sub> ratio has been proposed as a better indicator because HCHO and NO<sub>2</sub> have similar lifetimes and better represent the competition for OH radicals (Santiago et al., 2021; Tonnesen and Dennis, 2000).

O<sub>3</sub> formation occurs not only at the surface but also at elevated altitudes in the troposphere (Hu et al., 2024). Moreover, due to atmospheric convection, elevated O<sub>3</sub> can be dispersed downward to the near-surface layer. To investigate the vertical sensitivity of O<sub>3</sub> formation, some previous studies have employed column FNR observed by multi-axis differential optical absorption spectroscopy (MAX-DOAS) (Irie et al., 2021; Zhang et al., 2021; Ryan et al., 2023; Wang et al., 2025). Other studies have applied satellite-based techniques to assess the spatial sensitivity of O<sub>3</sub> formation (Duncan et al., 2010; Jin et al., 2017; Jung et al., 2022). By combining satellite and ground-based remote sensing, column FNR sheds light on the development of spatially and temporally targeted O<sub>3</sub> mitigation strategies.

The Pandora instrument is a passive UV-VIS spectrometer that observes solar photons over the 280–530 nm spectral range (Herman et al., 2009). The Pandonia Global Network (PGN) is a joint project supported by NASA and ESA, providing real-time, standardized, calibrated, and verified air quality data along with associated uncertainty estimates (<a href="https://www.pandonia-global-network.org/">https://www.pandonia-global-network.org/</a>). With more than 200 operational stations worldwide, the PGN has been widely applied in atmospheric research. In particular, due to their high accuracy, Pandora instruments have served as Fiducial Reference Measurement (FRM) for validating satellite observations (Douros et al., 2023; Judd et al., 2020; Kim et al., 2023) and airborne spectrometers (Choo et al., 2023; Judd et al., 2019). Additionally, Pandora observations have successfully highlighted the seasonal and diurnal variations of air pollutants (Herman and Mao, 2024; Liu et al., 2024). Mouat et al. (2024) reported a complex, heterogeneous environment near an airport using Pandora data. With its two viewing geometries, namely direct-sun and sky-scan modes, Pandora quantitatively observes total column amounts and lower tropospheric column amounts of several trace gases, including NO<sub>2</sub>, HCHO, and SO<sub>2</sub> (Cede et al., 2021a, b). Furthermore, the instrument provides both column and vertical distribution information, making it appropriate for investigating the O<sub>3</sub> formation sensitivity. Nevertheless, no studies have employed Pandora to analyze O<sub>3</sub> formation sensitivity.

Increased levels of air pollution not only influence health but also place a burden on socioeconomic costs and healthcare resources (Xu et al., 2025). Japan is facing the problem of an aging population that is more vulnerable to air pollution. A modeling study predicted that 80 % of Japan's population could be exposed to the highest levels of O<sub>3</sub> between 2030 and 2050 if no climate change mitigation policies are implemented (Chen et al., 2024). Looking back on history, the first episode of photochemical air pollution occurred in 1970, leading to the hospitalization of schoolchildren and high school students (Akimoto, 2017). In response, the government has implemented stringent emission controls since the 1980s, leading to a 56 % reduction in NO<sub>x</sub> and a 50 % reduction in VOCs emissions from 2000 to 2019 (Chatani et al., 2023). Despite the emission reductions, oxidant levels continued to rise. This paradox is potentially due to a decrease in NO titration and an increase in transboundary transport (Akimoto, 2017; Irie et al., 2021). Therefore, extensive studies on O<sub>3</sub> formation are needed to efficiently mitigate human exposure.

In this study, we first utilized both Pandora direct-sun and sky-scan modes to analyze O<sub>3</sub> formation sensitivity at different altitudes and latitudes across Japan. The Japan Pandora Network (JPN), as part of the PGN, has established more than 10

stations, providing real-time vertical measurements. Because FNR threshold values depend on the study region, meteorological conditions, and emissions, we applied the GEOS-Chem chemical transport model to determine region-specific FNR thresholds for Japan. To improve accuracy, we accounted for secondary HCHO contributions. The findings of this study provide scientific insights into the application of Pandora measurements for mitigating regional O<sub>3</sub> pollution.

#### 2. Methodology

#### 2.1 Surface measurements

To obtain surface concentrations of HCHO and NO<sub>2</sub>, we conducted in situ measurements at Tokyo Metropolitan University (Tokyo-TMU) during the summer (July 1–19) and fall (October 17–31) of 2022. NO<sub>2</sub> was measured using a cavity attenuated phase shift (CAPS) analyzer, which directly detects NO<sub>2</sub> by measuring absorption around 450 nm, with a detection limit of less than 0.1 ppb (Choi et al., 2020). Meanwhile, HCHO was obtained using a selected ion flow tube mass spectrometer (SIFT-MS). SIFT-MS utilizes precursor ions such as H<sub>3</sub>O<sup>+</sup>, NO<sup>+</sup>, and O<sub>2</sub><sup>+</sup> for ionization and detection of target substances (Langford et al., 2023; Roberts et al., 2022). This instrument is recommended as an efficient method for the measurement of HCHO in both indoor and outdoor environments (Zogka et al., 2022).

Surface  $O_3$  concentrations were obtained from nearby air monitoring stations using the UV absorption method. These stations are operated by the Atmospheric Environmental Regional Observation System (AEROS) (https://soramame.env.go.jp/).

# 2.2 Tropospheric column amounts and vertical profiles derived from Pandora observations

Pandora spectrometer consists of a head sensor, an optical fiber transmission system, and a charge-coupled device (CCD) used as a spectral detector. The data retrieval begins with the raw measurement spectra (L0). The corrected signal (L1) is obtained by applying complex corrections, such as dark correction, latency correction, etc. Spectral fitting (L2Fit) is performed to derive slant column densities relative to a reference spectrum using the differential optical absorption spectroscopy (DOAS) method. 105 Finally, L2 data is produced by converting slant columns into vertical columns utilizing geometrical air mass factors (AMFs) for the direct-sun mode and analytical methods for the sky-scan mode (Rawat et al., 2025). The direct-sun mode measures total NO<sub>2</sub> column with high precision (2.7×10<sup>14</sup> molecules cm<sup>-2</sup>) and accuracy (2.7×10<sup>15</sup> molecules cm<sup>-2</sup>). For total HCHO column, a statistical error of 6 % and a systematic error of 26 % have been reported (Spinei et al., 2018). The bias of the skyscan measurement is approximately  $-0.02 \times 10^{16}$  molecules cm<sup>-2</sup> for NO<sub>2</sub> and  $-0.05 \times 10^{16}$  molecules cm<sup>-2</sup> for HCHO (Tirpitz et al., 2021; Verhoelst et al., 2021). In this study, we explored both the tropospheric column FNR and vertical FNR profile by combining the direct-sun and sky-scan modes of Pandora. In the direct-sun mode, the tropospheric columns of NO2 and HCHO were derived by subtracting the stratospheric contribution. This stratospheric information was derived from the GEOS-Chem model (Sect. 2.3). In addition to column densities in the lower troposphere (up to an altitude of 3 km), the sky-scan measurements provided vertical distributions from the surface up to an altitude of 3 km with four layers of measurements. 115 These profiles are crucial for examining the vertical characteristics of NO2 and HCHO, as well as O3 production in the troposphere.

We used the Pandora data at four JPN stations, Sapporo, Tsukuba-NIES, Tokyo-TMU, and Fukuoka. These stations, listed from north to south, were chosen to investigate FNR across different latitude environments. A brief description of these Pandora stations is provided in Table S1 of the Supplementary Information. These Pandora instruments routinely alternate between direct-sun and sky-scan modes on a standard schedule (Cede et al., 2021a). Data processing was performed using the Blick software, which converts L0 (raw measurement spectra) to L2 products (e.g., vertical column densities, profiles, etc.). For FNR calculations, L2 products were employed. To maximize the available scientific data, we applied a new filtering method adopted from Rawat et al. (2025). The cut-off values were defined as the mean plus three standard deviations of the

independence uncertainty for data with a high-quality flag. Data with independence uncertainty exceeding the cut-off value was removed. We also excluded data with a solar zenith angle (SZA) > 75° (Mouat et al., 2024). Using this filtering method, the data volume increased significantly by 5-30 % for NO<sub>2</sub> and 20-70 % for HCHO, compared to the filtering method that uses high and medium data quality flags.

#### 2.3 Model simulations and determination of FNR thresholds

To characterize the FNR thresholds, we investigated the response of O<sub>3</sub> to emission perturbations using the GEOS-Chem model. GEOS-Chem is a three-dimensional chemical transport model driven by assimilated meteorological observations from the Goddard Earth Observing System (GEOS) of the NASA Global Modeling and Assimilation Office (GMAO) (http://www.geos-chem.org). To simulate O<sub>3</sub>, HCHO, and NO<sub>2</sub> for the year 2022, we used the high-performance GEOS-Chem (GCHP) model, version 14.4.0 (The International GEOS-Chem User Community, 2024). GCHP is described by Eastham et al. (2018). Improved advection, resolution, performance, and community access are described by Martin et al. (2022). In the 135 current study, we configured four model runs: a normal run (Run-1), a sensitivity run (Run-2), a reduced NO<sub>x</sub> emission run (Run-3), and a reduced VOCs emission run (Run-4).

For Run-1, global anthropogenic emissions were based on the Community Emissions Data System version 2 (CEDSv2) (McDuffie et al., 2020). We used the Regional Emission inventory in ASia version 3.2.1 (REASv3.2.1) (Kurokawa and Ohara, 2020), as the regional anthropogenic emissions override the global anthropogenic emissions for Japan. Biomass burning emissions were taken from the Global Fire Emissions Database version 4 (GFED4) (Van Der Werf et al., 2017). Additionally, dust, sea salt aerosol, soil NOx, lightning NOx, and biogenic VOCs emissions were computed offline (Weng et al., 2020). All emissions were configured at run-time using Harmonized Emissions Component (HEMCO, version 3.9.3) (Lin et al., 2021). Table S2 provides a detailed description of the emission inventories used in the model simulation. For meteorology, we used MERRA-2 (0.5° × 0.625°), a global atmospheric reanalysis data product. The full-chem model simulations use chemical mechanism kinetics following JPL/IUPAC recommendations (Bates et al., 2024). Photolysis frequencies for stratospheric and tropospheric chemistry are calculated with Cloud-J v7.7.3 (Prather, 2015). Stratospheric chemistry is represented by a linear chemistry mechanism, the Linoz algorithm (McLinden et al., 2000). The Linoz stratospheric chemistry package is recommended for GEOS-Chem simulations of O<sub>3</sub>. More details on the chemical mechanisms are available at http://www.geoschem.org. In our study, the simulations were run using a 10-minute time step for chemistry and a 5-minute time step for transport. Moreover, we applied the grid-stretching capability to focus on the Japan region. The grid-stretching procedure followed Bindle et al. (2021), with an initial cubed-sphere grid of C90, a target latitude of 37°, a target longitude of 137°, and a stretch factor of 4. This procedure yielded an average horizontal resolution of 27.78 km over Japan. The simulation generated the vertical extent from the surface to approximately 80 km with a 72 vertical-layer grid. Surface concentrations were obtained from the first model layer. The tropospheric column and the stratospheric column were separated using tropopause information. 155 Furthermore, this troposphere-stratosphere distribution was used to derive the tropospheric column from the Pandora directsun observations.

Run-2 was the same as Run-1 but with all anthropogenic HCHO emissions turned off. Secondary HCHO reflects the VOCs activity through photolysis reactions. Previous studies have highlighted the importance of separating secondary HCHO from anthropogenic HCHO for a more accurate interpretation of the FNR (Hong et al., 2022; Xing et al., 2022; Xue et al., 2022). By comparing Run-1 and Run-2, we determined the contribution of primary HCHO and excluded it from the FNR calculation. Run-3 and Run-4 were the same as Run-2 but with a 20 % reduction in regional NO<sub>x</sub> and VOC emissions, respectively. Ozone concentrations result from both in situ photochemical creation and external transport processes (Hong et al., 2022; Qian et al., 2024). A key advantage of the model-based method is that it allows us to exclude external transport processes, leading to a more precise classification of the O<sub>3</sub> chemical regime. The external O<sub>3</sub> transport influence was eliminated by subtracting Run-3 or Run-4 from Run-2. This step further reflects the response of O<sub>3</sub> to changes in NO<sub>x</sub> and VOCs. Figure S1 shows an https://doi.org/10.5194/egusphere-2025-5266 Preprint. Discussion started: 14 November 2025 © Author(s) 2025. CC BY 4.0 License.

175

190

195

example of the surface O<sub>3</sub> response to VOC and NO<sub>x</sub> emission reductions, resulting from GEOS-Chem simulations. Both negative and positive O<sub>3</sub> changes were observed in response to NO<sub>x</sub> emission reduction. In contrast, VOC emission reduction consistently led to decreases in O3 levels. The Greater Tokyo Metropolitan Area was strongly influenced by these emission perturbations.

Regarding the responses, the O3 sensitivity regime was categorized following the method of Jin et al. (2017) and Jung et al. (2022). A negative change in O3 owing to NOx emission reduction indicates a VOC-limited regime. A NOx-limited regime is defined when the positive change in O3 owing to VOC emission reduction is smaller than that from NOx emission reduction. The FNR threshold values for VOC-limited and NOx-limited regimes were determined as those corresponding to the 95th percentile of the cumulative probability distribution for each regime.

#### 3. Results and discussion

#### 3.1 Model simulations of HCHO, NO2, and O3

#### 3.1.1 Comparison with in situ and Pandora measurements

The correlation statistics between the GEOS-Chem simulations (Run-1) and in situ measurements are shown in Fig. S2, and those with Pandora are shown in Fig. S3. The GEOS-Chem simulations underestimated the surface HCHO concentrations, with a slope of 0.37, a correlation coefficient (R) of 0.32, and a root mean square error (RMSE) of 1.56 ppbv. Meanwhile, the GEOS-Chem overestimated the surface NO<sub>2</sub> concentrations, with a slope of 1.34, R = 0.48, and RMSE = 14.65 ppbv. However, the GEOS-Chem simulations aligned better with the Pandora tropospheric column densities. Specifically, the correlation coefficients between GEOS-Chem and Pandora for HCHO and NO2 were 0.51 and 0.56, respectively, for Sapporo, 0.87 and 0.72 for Tsukuba-NIES, 0.78 and 0.54 for Tokyo-TMU, and 0.69 and 0.70 for Fukuoka. The RMSE varied from 2.99 to 5.93 185 Pmolec cm<sup>-2</sup> for HCHO and from 2.11 to 5.71 Pmolec cm<sup>-2</sup> for NO<sub>2</sub>. The surface conditions are likely more complex compared to the tropospheric column, which could explain why the GEOS-Chem model imperfectly captured the surface characteristics. Additionally, the local emission inventory implemented in the model was based on the year 2015, which might cause a significant bias in the model results. The diurnal cycles of HCHO and NO2 simulated by the GEOS-Chem model were compared with in situ measurements (Fig. S4) and Pandora (Fig. S5). The GEOS-Chem model successfully reproduced the surface diurnal cycle of HCHO. For surface NO2, the simulation captured the diurnal cycle in October quite well, but dramatically overestimated NO<sub>2</sub> concentrations in July. Both the Pandora observations and the GEOS-Chem simulations showed a midday decrease in tropospheric NO2 columns. The GEOS-Chem model also simulated the growth in HCHO from the morning; however, it showed a slight decrease after 2:00 PM. A larger difference between the model and Pandora was observed at Tokyo-TMU, where primary HCHO emissions are significant.

For surface O<sub>3</sub>, we compared the GEOS-Chem model performance with in situ measurements from nearby air monitoring stations at the four Pandora locations. The GEOS-Chem simulation was able to reproduce the inverted U-shaped pattern of O<sub>3</sub> but showed positive biases (15-25 ppbv) (Fig. 1). The O<sub>3</sub> depletion during nighttime was not well generated by the GEOS-Chem model, resulting in overestimated concentrations during photochemical periods. The faster nighttime depletion rate of O<sub>3</sub> at Tokyo-TMU and Tsukuba-NIES suggests stronger NO titration, which was not reflected by the model simulation. High positive biases in surface O<sub>3</sub> have been reported in the literature (Travis et al., 2016; Travis and Jacob, 2019). The diurnal variations in mixed-layer dynamics and ozone deposition velocities in the model are one of the key factors contributing to this bias (Travis and Jacob, 2019). However, GEOS-Chem generally captured the observed variations in daytime O<sub>3</sub>, such as large fluctuations at Tokyo-TMU and Tsukuba-NIES, and narrower variations at Fukuoka and Sapporo. The agreement between model simulation and in situ measurements was moderate, with an R of 0.56, 0.61, 0.53, and 0.59 for Sapporo, Tsukuba-NIES,

Tokyo-TMU, and Fukuoka, respectively. Overall, the GEOS-Chem model could capture O<sub>3</sub> production at the study locations, indicating its suitability for investigating the O<sub>3</sub> sensitivity regime.

Figure 1: Comparison of hourly surface  $O_3$  between the GEOS-Chem model and in situ measurements at the study locations. The top-row panels show scatterplots with linear regression equations. The bottom-row panels present the diurnal cycles of surface  $O_3$ , with shaded error bands indicating  $\pm 1$  standard error.

#### 3.1.2 Contribution of secondary HCHO

Table 1 shows the contributions of secondary HCHO at the study locations, derived from the GEOS-Chem simulations. We only considered daytime simulation from 8:00 to 16:00, when photochemical reactions actively occur. Basically, secondary HCHO contributed the majority at the surface level, with values of 86 %, 87 %, 76 %, and 90 % at Sapporo, Tsukuba-NIES, Tokyo-TMU, and Fukuoka, respectively. The contribution of primary HCHO was slightly higher at Tokyo-TMU (24 %). The secondary HCHO contribution increased in summer and decreased in winter. A diurnal variation of the contribution of secondary HCHO was observed, with the highest contribution around noon (not shown). For the tropospheric column, the seasonal contribution of secondary HCHO remained consistent. Additionally, the secondary HCHO contribution increased with altitude (Fig. S6). The contribution at altitudes above 3 km was almost entirely attributed to secondary formation.

Secondary HCHO effectively represents a proxy for VOC reactivity (Su et al., 2019; Xue et al., 2022). For more accurate FNR calculation, these secondary HCHO contributions from the model simulations were adopted for both in situ and Pandora measurements. The FNR using secondary HCHO is referred to as FNR<sub>sec</sub>. This approach is particularly important in areas with a high contribution of primary sources of HCHO, for example, urban and industrial areas.

Table 1: Contribution of secondary HCHO (%) at the surface level and in the tropospheric columns, obtained from the GEOS-Chem model simulation at the JPN sites. The statistical values were considered only for 8-hour daytime simulations (8:00 to 16:00).

| Location     |                     | Spring      | Summer     | Fall      | Winter      | Annual average |
|--------------|---------------------|-------------|------------|-----------|-------------|----------------|
| Sapporo      | Surface             | $89 \pm 13$ | $98 \pm 1$ | 91 ± 9    | $65 \pm 15$ | 86 ± 17        |
|              | Tropospheric column | $98\pm2$    | $99\pm0$   | $98\pm2$  | $93\pm3$    | $97\pm3$       |
| Tsukuba-NIES | Surface             | $89\pm11$   | $96\pm4$   | $91\pm10$ | $74\pm16$   | $87\pm14$      |
|              | Tropospheric column | $96 \pm 4$  | $99\pm1$   | $97\pm3$  | $90\pm 6$   | $95 \pm 5$     |
| Tokyo-TMU    | Surface             | $76\pm16$   | $92\pm 8$  | $77\pm19$ | $58 \pm 22$ | $76\pm21$      |
|              | Tropospheric column | $91\pm 5$   | $97\pm2$   | $93\pm 5$ | $85\pm 8$   | $92 \pm 7$     |
| Fukuoka      | Surface             | $90 \pm 9$  | $97\pm3$   | $92\pm 8$ | $79\pm13$   | $90 \pm 11$    |
|              | Tropospheric column | $97\pm3$    | $99\pm0$   | $98\pm2$  | $94 \pm 3$  | $97\pm3$       |

#### 3.2 Overall levels of HCHO and NO2

#### 3.2.1 Surface levels

The diurnal plots of HCHO and NO<sub>2</sub> obtained from in situ measurements at the Tokyo-TMU site are shown in Fig. S4. Near the surface, HCHO concentrations fluctuated within a narrow range of a few ppbv throughout the day. The minimum occurred at mid night, while the peak was observed around noon (in July) or late afternoon (in October). HCHO concentrations are generally expected to be higher in summer due to enhanced photochemical reactions driven by stronger solar irradiance and increased biogenic VOCs emissions from the local flora (Irie et al., 2021). However, interestingly, we found negligible differences in surface HCHO concentrations between July and October. The mean HCHO concentrations in July and October were  $1.82 \pm 1.14$  and  $2.02 \pm 1.38$  ppbv, respectively.

Surface NO<sub>2</sub> concentrations were twice as high in October compared to July. In July, the NO<sub>2</sub> mixing ratio ranged from 0.90 to 21.10 ppbv, with an average of 5.98 ppbv. In October, the surface NO<sub>2</sub> ranged from 1.24 to 39.13 ppbv, with an average of 9.91 ppbv. Peaks in July were not clearly defined. In contrast, October showed a small peak in the morning and a larger one around 18:00, exhibiting traffic emissions, while minimum concentrations occurred at noon due to photochemical loss. The finding that the diurnal trend of NO<sub>2</sub> was opposite to that of HCHO can be explained by photochemical reactions. These diurnal cycles have been well documented in previous studies (Irie et al., 2011; Itahashi and Irie, 2022).

#### 3.2.2 Vertical column amounts

Figure 2 shows the variation in vertical column density (VCD) of HCHO measured by the Pandora spectrometer in 2022 at the study sites. The highest total VCD was observed at Tokyo-TMU, with a mean value of 12.30 ± 4.99 Pmolec cm<sup>-2</sup>. The levels at Sapporo, Tsukuba-NIES, and Fukuoka were 6.01 ± 2.84, 8.97 ± 5.05, and 9.05 ± 4.02 Pmolec cm<sup>-2</sup>, respectively.

According to the GEOS-Chem simulations, the contribution of primary HCHO at Tokyo-TMU was higher than that at the other study sites (Table 1), indicating that the elevated column amount at Tokyo-TMU was largely influenced by local emission of anthropogenic sources. Clearly, the HCHO column amounts tended to be higher in summer and lower in winter. As discussed in Sect. 3.2.1, the difference in surface HCHO concentrations between July and October at Tokyo-TMU was not statistically significant. However, the total HCHO VCD measured by Pandora in July (16.47 ± 5.05 Pmolec cm<sup>-2</sup>) was 1.5

times higher than that in October (10.90 ± 2.05 Pmolec cm<sup>-2</sup>). Additionally, we found a strong relationship between column densities and surface concentrations, with a correlation of 0.7 for HCHO (Fig. S7). The correlation slope was about three times higher in July than in October, likely due to enhanced HCHO production at higher altitudes during summer. This suggests that surface measurements alone may not fully represent atmospheric HCHO. Therefore, column information is essential.

The lower tropospheric column (up to an altitude of 3 km) of HCHO (obtained from the sky-scan mode) accounted for 59.14 ± 18.83 % at Sapporo, 65.86 ± 14.84 % at Tsukuba-NIES, 49.04 ± 16.76 % at Tokyo-TMU, and 62.27 ± 16.66 % at Fukuoka, relative to the total column amount (obtained from the direct-sun mode). This lower tropospheric contribution was dominant in summer, ranging from 61.27 % to 71.65 %, and decreased in winter, varying from 32.92 % to 54.93 %. The diurnal cycle of the tropospheric HCHO column is shown in Fig. S5. The tropospheric column measured by Pandora exhibited a continuous increase from the morning to the afternoon. The accumulation of HCHO and the decreasing rate of photolysis in the afternoon can explain this daily cycle of HCHO (Zhang et al., 2021).

In contrast to the HCHO trend, the NO<sub>2</sub> column amounts reached their minimum in summer (Fig. 2). Tokyo-TMU was also polluted with NO<sub>2</sub>, with an average of 9.73 ± 4.99 Pmolec cm<sup>-2</sup> (12.55 ± 6.52 Pmolec cm<sup>-2</sup> in winter and 6.80 ± 2.17 Pmolec cm<sup>-2</sup> in summer), followed by Tsukuba-NIES with an average of 8.33 ± 3.88 Pmolec cm<sup>-2</sup> (8.83 ± 4.42 Pmolec cm<sup>-2</sup> in winter and 7.84 ± 2.50 Pmolec cm<sup>-2</sup> in summer). The total NO<sub>2</sub> VCDs at Sapporo and Fukuoka were 7.92 ± 4.45 Pmolec cm<sup>-2</sup> and 7.22 ± 3.01 Pmolec cm<sup>-2</sup>, respectively. The lower tropospheric column contributed 51.38 ± 20.86 %, 53.95 ± 18.06 %, 59.55 ± 16.79 %, and 56.29 ± 18.83 % to the total column at Sapporo, Tsukuba-NIES, Tokyo-TMU, and Fukuoka,

respectively. We also found good agreement for NO<sub>2</sub> between lower tropospheric column densities and surface concentrations, with a correlation of 0.6 (Fig. S7). The slope was similar in July and October, reflecting that NO<sub>2</sub> is mainly concentrated near the surface.

Figure 2: Time series plots of vertical column densities of HCHO (left column) and  $NO_2$  (right column) at the study locations in 2022. Black plus signs represent total column densities derived from the Pandora direct-sun mode, whereas blue dots indicate lower tropospheric columns (up to an altitude of 3 km) derived from the Pandora sky-scan mode. Solid lines show monthly means, and error bars represent standard deviation. (1 Pmolec cm<sup>-2</sup> =  $1 \times 10^{15}$  molecules cm<sup>-2</sup>).

#### 3.2.3 Vertical profiles

The Pandora sky-scan mode measures scattered sunlight at several angles, yielding the vertical profiles. The resolution of the vertical profiles depends on the number of scanning angles, referred to as elevation scan routines (i.e., detailed elevation scan and quick elevation scan). Typically, the four Pandora stations operated using the quick elevation scan, which yields four partial vertical columns of the lower troposphere. We converted these partial vertical columns to mixing ratios, assuming well-mixed conditions in each layer and the ideal gas law. To obtain the diurnal and seasonal vertical distributions of HCHO and NO2 up to 3 km altitude, after extrapolating into 0.1 km bins, 7337–9201 NO2 profiles and 11760–15651 HCHO profiles at each station were averaged on an hourly basis. Since the altitudes of these four Pandora stations ranged from 45 to 135 m, we only considered the vertical profile from 0.2 to 3 km. Figure 3 presents the diurnal and seasonal profiles of HCHO and NO2 at Tsukuba-NIES, with the planetary boundary layer height (PBLH) derived from MERRA-2. The profiles for the Tokyo-TMU, Sapporo, and Fukuoka sites are shown in Fig. S8.

Figure 3: Vertical profiles of HCHO (top panels) and NO<sub>2</sub> (bottom panels) derived from the Pandora sky-scan mode at Tsukuba-NIES. The leftmost panels show the seasonal vertical profiles, with shaded error bands indicating ±1 standard error. Color bars exhibit diurnal profiles. Black dashed curves present planetary boundary layer height (PBLH).

The HCHO molecules were significantly enhanced at noon during summer due to stronger solar intensity and increased biogenic VOCs emissions such as isoprene. Secondary HCHO is associated with isoprene emitted from vegetation (Ryan et al., 2023). Isoprene emissions exhibit a markedly positive exponential relationship with temperature (Ryan et al., 2023; Wang et al., 2024). At higher temperatures in summer, more isoprene is emitted, which triggers the formation of secondary HCHO. In the current study, higher amounts of HCHO commonly occurred in the late afternoon in spring and fall. The vertical variation of HCHO appeared to be relatively heterogeneous in winter. However, we found less significant changes in the diurnal and seasonal profiles at the highest latitude site (Sapporo). Vertically, HCHO formation was observed up to 2 km, which is consistent with previous studies (Lin et al., 2022; Shi et al., 2025b). HCHO extended beyond the PBLH, suggesting an important contribution from oxidation of VOCs at higher altitudes. Surface measurements of HCHO were unable to capture its characteristic in the upper layers. Hong et al. (2022) reported that HCHO concentrations below 0.2 km remain consistent on both non-exceedance and polluted days. Therefore, surface measurements likely miss information on VOC oxidation processes occurring aloft. For a more accurate FNR analysis, it is essential to consider not only surface but also elevated measurements.

The NO<sub>2</sub> distributions rapidly decreased with increasing altitude. Unlike HCHO, the bulk of NO<sub>2</sub> was generally concentrated below PBLH because NO<sub>x</sub> emissions are mainly near the surface (i.e., traffic). Consequently, the diurnal trend followed a pattern similar to that at the surface, which exhibited a peak in the morning, a reduction during noon, and an increase during the afternoon rush hours. Photochemical loss of NO<sub>2</sub> during noon was less intense in winter than in summer. In winter, the low boundary layer height and reduced solar irradiance led to NO<sub>2</sub> accumulation near the surface. At the Sapporo site, NO<sub>2</sub> levels were significantly enhanced in winter (Fig. S8). PBLH sank to below 1 km throughout the day, resulting in the accumulation of emissions.

In addition to meteorological conditions, the vertical profiles of HCHO and NO<sub>2</sub> are influenced by both local initial emissions and external pollution transport (Shi et al., 2025a). To investigate whether any external transport affected the vertical profiles, we applied the Hybrid Single-Particle Lagrangian Integrated Trajectory (HYSPLIT) model for the Tsukuba-NIES case. The model, developed by the National Oceanic and Atmospheric Administration (NOAA) Air Resources Laboratory (ARL), has been widely used for atmospheric trajectory and dispersion calculations (Stein et al., 2015). We simulated 48-hour backward trajectories for the year 2022, with endpoints at 13:00 local time (JST) at Tsukuba-NIES, as a case study. The

backward trajectories were assigned into four clusters: c1, c2, c3, and c4. The number of clusters was optimized using total spatial variance (TSV) analysis. The cluster frequencies were 15 %, 40 %, 36 %, and 9 % for c1, c2, c3, and c4, respectively (Fig. S9). Figure 4 shows the vertical profiles of HCHO and NO2 at Tsukuba-NIES as a function of air mass clusters. Cluster c2, which occurred most frequently throughout the year, generally exhibited higher concentrations of both HCHO and NO2. Clusters c3 and c1 showed lower concentrations. Cluster c1 was associated with transboundary air mass from North China through the Sea of Japan, while cluster c3 originated around the eastern coastline. These oceanic air masses likely diluted the pollution. In contrast, cluster c4, which passed through urban Tokyo and industrial areas, transported anthropogenic pollutants to the Tsukuba-NIES site. Notably, concentrations of HCHO and NO2 were slightly enhanced during occurrences of cluster c4, even in summer months (July and August). This suggests that external pollution transport affected the vertical distributions of both HCHO and NO2, and consequently, O3 production.

Figure 4: Clustered vertical profiles of HCHO (top panels) and NO<sub>2</sub> (bottom panels) at Tsukuba-NIES. Shaded error bands indicate ±1 standard error. Color bars exhibit diurnal profiles.

#### 3.3 Identification of the O<sub>3</sub> sensitivity regime

The  $O_3$ , HCHO, and  $NO_2$  outputs from the GEOS-Chem model simulations were utilized to identify the  $O_3$  sensitivity regime (Fig. 5). From the scatter plots, the responses of surface  $O_3$  to  $NO_x$  and VOC emission reductions at Sapporo and Fukuoka were within 5 ppbv, which were less pronounced than those at Tsukuba-NIES and Tokyo-TMU (within 10 ppbv). A positive  $O_3$  difference indicates that the emission reduction results in a decrease in surface  $O_3$ . Conversely, a negative difference means that the emission reduction enhances surface  $O_3$  concentrations. In our study, generally, VOC emission reductions led to a decrease in surface  $O_3$  concentrations.  $NO_x$  emission reductions could either decrease  $O_3$  through photochemical reaction or increase it due to the NO titration effect.

The FNR<sub>sec</sub> threshold values were determined using both surface and column information (Fig. 5, cumulative graphs). Based on surface FNR<sub>sec</sub>, the VOC-limited conditions were distinguished with thresholds of FNR<sub>sec</sub> <0.2 for Sapporo, <0.29 for Tsukuba-NIES, <0.28 for Tokyo-TMU, and <0.18 for Fukuoka. The NO<sub>x</sub>-limited regimes were associated with FNR<sub>sec</sub> >0.36 for Sapporo, and >0.75, and >0.75 and >0.72 for Tsukuba-NIES, Tokyo-TMU, and Fukuoka, respectively. The surface FNR<sub>sec</sub> thresholds were generally lower than the column FNR<sub>sec</sub>. For instance, the column FNR<sub>sec</sub> thresholds for the VOC-limited regimes and NO<sub>x</sub>-limited regimes at the Sapporo site were <0.86 and >1.52, respectively. The corresponding column FNR<sub>sec</sub> threshold ranges for Tsukuba-NIES, Tokyo-TMU, and Fukuoka were 0.74–1.7, 0.62–1.53, and 0.43–1.73, respectively.

The transitional regime seemed to occur over a wider range of column FNR<sub>sec</sub> values at lower latitudes (e.g., Fukuoka) compared to higher latitudes (e.g., Sapporo). The differences between surface and tropospheric column FNR<sub>sec</sub> could be attributed to the vertical characteristics of HCHO and NO<sub>2</sub>. NO<sub>2</sub> molecules were concentrated near the surface, while HCHO was present in the higher layers. As a result, surface FNR<sub>sec</sub> did not account for HCHO at higher layers, leading to lower threshold values. The surface and column FNR<sub>sec</sub> thresholds identified in this study are consistent with those reported by Jin et al. (2017) for East Asia (Table 2).

Figure 5: Response of surface O<sub>3</sub> to NO<sub>x</sub> and VOC emission perturbations resulting from the GEOS-Chem simulations. The scatter plots depict the O<sub>3</sub> difference between GEOS-Chem Run-2 and Run-3 (blue), and between Run-2 and Run-4 (brown), as a function of surface FNR<sub>sec</sub> (first row) and tropospheric column FNR<sub>sec</sub> (second row). The VOC-limited regime was associated with negative change in O<sub>3</sub> due to NO<sub>x</sub> emission reduction. The NO<sub>x</sub>-limited regime was identified when the positive change in O<sub>3</sub> due to VOC emission reductions was smaller than that from NO<sub>x</sub> emission reductions. Line plots show the cumulative probability of NO<sub>x</sub>-limited (blue) and VOC-limited (brown) conditions as a function of surface FNR<sub>sec</sub> (third row) and tropospheric column FNR<sub>sec</sub> (fourth row). The FNR<sub>sec</sub> threshold values (vertical dashed lines) for VOC-limited and NO<sub>x</sub>-limited regimes were determined as those corresponding to the 95<sup>th</sup> percentile (horizontal dashed lines) of the cumulative probability distribution for each regime.

Table 2 presents the column FNR regime thresholds related to surface O<sub>3</sub> sensitivity from previous studies. These threshold values vary depending on the methodology, geographic region, and atmospheric conditions. The FNR thresholds using both

primary and secondary HCHO (FNR<sub>total</sub>) are higher than those using only secondary HCHO (FNR<sub>sec</sub>). The transitional regimes are reported for a column FNR<sub>total</sub> of 2.5–4.0 in Guangzhou, China (Hong et al., 2022), and 1.6–2.6 in United States (Jung et al., 2022), whereas the column FNR<sub>sec</sub> thresholds identified in our study were lower than 2. Column FNR regime threshold values are useful for examining global O<sub>3</sub> production, as both satellite and ground-based remote sensing techniques offer extensive spatial coverage (Inoue et al., 2019; Ryan et al., 2023; Santiago et al., 2021).

Table 2: Comparison of column FNR threshold values for O<sub>3</sub> sensitivity in previous studies using different methods.

| Study area       | Indicator                   | Method                           | FNR <sub>total</sub> threshold values | FNR <sub>sec</sub> threshold values | Reference           |
|------------------|-----------------------------|----------------------------------|---------------------------------------|-------------------------------------|---------------------|
| North America    | Column FNR <sub>total</sub> | GEOS-Chem model                  | 0.9-1.4                               | -                                   | (Jin et al., 2017)  |
| Europe           | $Column\;FNR_{total}$       | GEOS-Chem model                  | 0.9-1.2                               | -                                   | (Jin et al., 2017)  |
| East Asia        | Column FNR <sub>total</sub> | GEOS-Chem mode                   | 0.9-1.6                               | -                                   | (Jin et al., 2017)  |
| United States    | Column FNR <sub>total</sub> | CMAQ model                       | 1.6-2.6                               | -                                   | (Jung et al., 2022) |
| Guangzhou, China | Column FNR <sub>total</sub> | Polynomial fit of O <sub>3</sub> | 2.5-4.0                               | -                                   | (Hong et al., 2022) |
| Sapporo          | Column FNR <sub>sec</sub>   | GEOS-Chem model                  | -                                     | 0.86-1.52                           | This study          |
| Tsukuba-NIES     | Column FNR <sub>sec</sub>   | GEOS-Chem model                  | -                                     | 0.74-1.70                           | This study          |
| Tokyo-TMU        | Column FNR <sub>sec</sub>   | GEOS-Chem model                  | -                                     | 0.62-1.53                           | This study          |
| Fukuoka          | Column FNR <sub>sec</sub>   | GEOS-Chem model                  | -                                     | 0.43-1.73                           | This study          |

#### 3.4 Analysis of O<sub>3</sub> sensitivity using in situ and ground-based remote sensing measurements

# 370 3.4.1 Surface FNR<sub>sec</sub>

Based on the surface FNR<sub>sec</sub> threshold values of 0.28–0.75 identified for Tokyo-TMU, we analyzed surface O<sub>3</sub> formation sensitivity using in situ measurements (Fig. 6). Surface FNR<sub>sec</sub> denotes the ratio of surface secondary HCHO to NO<sub>2</sub>. The surface secondary HCHO was derived from assimilated in situ measurements and GEOS-Chem model simulations. Surface FNR<sub>sec</sub> developed in the early morning and decreased in the afternoon. During the 8-hour daytime period (8:00 to 16:00), the average surface FNR<sub>sec</sub> was 0.65 ± 0.30 in July and 0.47 ± 0.42 in October. Transitional regimes accounted for 49 % and 31 % of O<sub>3</sub> chemical regimes in July and October, respectively. In July, VOC-limited and NO<sub>x</sub>-limited regimes contributed 11% and 40 %, respectively, while in October, they contributed 47 % and 22 %, respectively. The surface layer was predominantly in VOC-limited regimes in the morning, driven by NO<sub>x</sub> emissions from traffic. Photochemical reactions at noontime greatly shifted the O<sub>3</sub> formation sensitivities toward NO<sub>x</sub>-limited regimes. The NO depletion strongly triggered the O<sub>3</sub> chemical regime transition (Sakamoto et al., 2019). The NO<sub>x</sub>-limited regime occurred for a shorter time in October (12:00 – 14:00) than in July (10:00 – 14:00).

Figure 6: Variation in the surface FNR<sub>sec</sub> at Tokyo-TMU. The left panel shows the diurnal cycle. Shaded error bands indicate ±1 standard error. The box plots indicate monthly FNR<sub>sec</sub> with the median (solid line) and mean (triangle). Horizontal dashed lines present the surface FNR<sub>sec</sub> thresholds.

#### 3.4.2 Tropospheric column FNR<sub>sec</sub>

The tropospheric column FNR<sub>sec</sub> variation is shown in Fig. 7. Using vertical tropospheric column densities, the column FNR<sub>sec</sub> exhibited a higher value compared to those used in surface measurements. Like surface FNR<sub>sec</sub>, the tropospheric column FNR<sub>sec</sub> grew during the morning and peaked around noon. The highest column FNR<sub>sec</sub> values were found during the summer months (JJA), reaching  $3.59 \pm 2.10$  for Sapporo,  $3.29 \pm 1.26$  for Tsukuba-NIES,  $4.04 \pm 1.47$  for Tokyo-TMU, and  $5.29 \pm 3.20$  for Fukuoka. In contrast, these values were normally below 1 in winter (DJF). Based on the tropospheric column FNR<sub>sec</sub>, the O<sub>3</sub> formation mainly fell into the NO<sub>x</sub>-limited regime during summer and fall (SON). NO<sub>x</sub>-limited conditions accounted for 72 %, 46 %, 50 %, and 57 % of the chemical regime for Sapporo, Tsukuba-NIES, Tokyo-TMU, and Fukuoka, respectively. This regime was dominant in summer, with contributions exceeding 80%. The high temperatures, increased VOCs emissions, and strong photochemical activity during summer led to the dominance of NO<sub>x</sub>-limited conditions. The transitional regime accounted for 17 %, 41 %, 35 %, and 39 % for Sapporo, Tsukuba-NIES, Tokyo-TMU, and Fukuoka, respectively. The O<sub>3</sub> production was mainly sensitive to both NO<sub>x</sub> and VOC in winter. The VOC-limited regime occurred less frequently, varying from 4 % to 15 %, and was only observed in winter and spring (MAM).

Figure 7: Tropospheric column FNR<sub>sec</sub> obtained from Pandora measurements at the study locations. The left columns show the diurnal cycle. Shaded error bands indicate  $\pm 1$  standard error. The box plots indicate monthly column FNR<sub>sec</sub> with the median (solid line) and mean (triangle).

https://doi.org/10.5194/egusphere-2025-5266 Preprint. Discussion started: 14 November 2025 © Author(s) 2025. CC BY 4.0 License.

# 3.4.3 Vertical FNR<sub>sec</sub> profiles

Figure 8 illustrates the vertical distribution of FNR<sub>sec</sub> and the corresponding O<sub>3</sub> sensitivity regime. We observed an elevated trend of FNR<sub>sec</sub> from the surface, which peaked at around 1.5 km for Tsukuba-NIES, Tokyo-TMU, and Fukuoka. The range of FNR<sub>sec</sub> widened between 1 and 2 km altitude. In Sapporo, FNR<sub>sec</sub> kept increasing with altitude without a clear peak. This vertical trend of FNR<sub>sec</sub> was consistent across seasons; however, the largest magnitudes were found in summer, followed by fall, spring, and winter. During winter, the vertical FNR<sub>sec</sub> values remained below 1.5. In the summertime, the FNR<sub>sec</sub> profiles varied from 1 to 4 for Sapporo, 1 to 5 for Tsukuba-NIES and Tokyo-TMU, and 1 to 6 for Fukuoka.

The transition among O<sub>3</sub> production regimes depended on season, time of day, and altitude. Overall, the seasonal vertical distribution of the O<sub>3</sub> sensitivity regimes showed more NO<sub>x</sub>-limited regimes in summer, more transitional regimes in spring and fall, and more VOC-limited regimes in winter (Fig. 8). With increasing altitude, VOC-limited conditions typically formed near the surface layers, followed by transitional regimes in the mid-levels, and NO<sub>x</sub>-limited regimes aloft. The vertical sensitivity regime had a slight difference across the study locations. The VOC-limited regime appeared to have a higher probability at higher latitudes. In Sapporo, VOC-limited regimes occurred in all four seasons and were normally confined to below 1 km in spring, summer, and fall. In winter, the VOC-limited conditions extended throughout the daytime and up to 3 km altitude. The NO<sub>x</sub>-limited regime dominated at higher altitudes, expanding downward to the surface during spring and summer, and retreating in fall and winter.

Figure 8: Vertical FNR<sub>sec</sub> and ozone sensitivity regime contributions. Solid lines show the seasonal FNR<sub>sec</sub>, with shaded error bands indicating  $\pm 1$  standard error. Color bars indicate the NO<sub>x</sub>-limited regime (orange), transitional regime (green), and VOC-limited regime (blue). The specific FNR<sub>sec</sub> thresholds for each location are shown in parentheses.

#### Conclusions

This study demonstrates the use of Pandora measurements to investigate O<sub>3</sub> sensitivity. Pandora provides vertical column densities and profiles of NO<sub>2</sub> and HCHO, which are valuable for a comprehensive understanding of O<sub>3</sub> production in the lower to mid-troposphere. Additionally, with the aid of the GEOS-Chem chemical transport model, we identified the region-specific thresholds to improve sensitivity analysis.

By applying the grid-stretching capability of the high-performance GEOS-Chem model, the diurnal cycles of NO<sub>2</sub>, HCHO, and O<sub>3</sub> were generally well reproduced. The correlation coefficients between the model and in situ surface measurements were 0.48 for NO<sub>2</sub> and 0.32 for HCHO, and from 0.53 to 0.61 for O<sub>3</sub>. For the tropospheric columns, the correlations between GEOS-Chem and Pandora ranged from 0.54 to 0.72 for NO<sub>2</sub>, and from 0.51 to 0.87 for HCHO.

According to the GEOS-Chem simulations, secondary HCHO was the dominant contributor to total HCHO, with its contribution increasing with altitude. Furthermore, the region-specific thresholds for the O<sub>3</sub> sensitivity regime were identified. The surface FNR<sub>sec</sub> values tended to be lower compared to the column FNR<sub>sec</sub>. The column FNR<sub>sec</sub> threshold ranges were 0.86–1.52 in Sapporo, 0.74–1.7 in Tsukuba-NIES, 0.62–1.53 in Tokyo-TMU, and 0.43–1.73 in Fukuoka.

Based on the surface FNR<sub>sec</sub>, the  $O_3$  sensitivity analysis revealed that the surface layer was predominantly in a VOC-limited regime in the morning, driven by traffic-related NO<sub>x</sub> emissions. Photochemical reactions shifted the  $O_3$  formation regime toward NO<sub>x</sub>-limited conditions at noon. Using the tropospheric column FNR<sub>sec</sub>, the  $O_3$  formation regime was found to be mainly NO<sub>x</sub>-limited during summer and fall, due to increasing VOCs emissions and strong photochemical activity. In winter,  $O_3$  production was sensitive to both NO<sub>x</sub> and VOCs. Meanwhile, the VOC-limited regime occurred less frequently and was observed in winter and spring. The vertical distribution of  $O_3$  sensitivity regimes was also obtained from the Pandora vertical profile.

To mitigate O<sub>3</sub> exposure, particularly for sensitive groups, policymakers should prioritize VOC emission controls near the surface layers in higher-latitude locations. During summertime, greater attention should be paid to controlling local NO<sub>x</sub> emissions. Moreover, regional transport of NO<sub>x</sub> from large emission sources can contribute to elevated NO<sub>x</sub> layers, where the NO<sub>x</sub>-limited regime is dominant, thereby enhancing O<sub>3</sub> production.

### Data availability

The Pandora data are available at the PGN website (https://www.pandonia-global-network.org).

#### 450 Author contributions

NC: Conceptualization, Formal analysis, Writing (original draft preparation). HT: Conceptualization, Supervision, Writing (review and editing). SI: Conceptualization, Writing (review and editing). TF, MF, SK, and HT: Data curation, Writing (review and editing). KI, YD, and RU: Writing (review and editing).

# **Competing interests**

The authors declare that they have no conflict of interest.

# Acknowledgements

We thank the principal investigators, support staff, and funding for establishing and maintaining the Sapporo, Tsukuba-NIES, Tokyo-TMU, and Fukuoka sites of the PGN used in this investigation. The PGN is a bilateral project supported by funding from NASA and ESA. The model simulations were run using the NIES scalar processing supercomputer (HPE Apollo 2000).

We thank Ms. Kimiko Suto (NIES) for managing the air monitoring data. This work was financially supported by JST SPRING, Grant Number JPMJSP2125. The author (N.D.T.C) would like to take this opportunity to thank the "THERS Make New Standards Program for the Next Generation Researchers".

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
