# Peer review of "Ozone formation sensitivity based on the secondary formaldehyde-tonitrogen\_dioxide ratio (FNRsec) derived from ground-based remote sensing measurements and a chemical transport model"

_EGUsphere, 2025_

## Referee Comment (RC2)

**General comments**

This manuscript presents a comprehensive analysis of ozone ($O_3$) formation sensitivity across Japan using the secondary formaldehyde-to-nitrogen dioxide ratio (FNRsec), with thresholds derived from GEOS-Chem chemical transport model simulations and sensitivity derived from combination of ground-based observations. The study showed that the vertical and temporal variability of ozone sensitivity is critical for designing effective mitigation strategies.

The main strengths of the manuscript include (1) the novel application of Pandora direct-sun and sky-scan measurements to investigate $O_3$ sensitivity regimes (2) the explicit separation of secondary and primary HCHO using model-based diagnostics. Overall, the observational–model framework is well designed, and the analysis is detailed.

However, several aspects require clarification or further justification before the conclusions can be fully supported. In particular: (1) the uncertainty and limitations associated with using space-based or column-based FNR indicators to infer $PO_3$ regimes should be discussed (2) the importance of understanding vertical distribution of ozone production sensitivity should be emphasized more clearly in the introduction to better motivate the novelty of the work (3) the reliability of the FNRsec thresholds depends strongly on model performance for HCHO and $NO_2$, and primary and secondary HCHO separation, which need further discussion.

Overall, the current form of manuscript requires **major revision** before it can be considered suitable for publication. Addressing the scientific and methodological issues outlined below would substantially strengthen the study.

**Specific comments**

1. **Consistency between FNRsec thresholds and observational application.**

   It is somewhat confusing that the FNRsec thresholds are derived from GEOS-Chem perturbation simulations using secondary HCHO, but the ratios for in-situ and Pandora observations are total HCHO/$NO_2$ (hereafter FNR_all). Applying the thresholds to the observations to diagnose the $PO_3$ sensitivity, which means comparing FNR_all with FNRsec-based thresholds, may introduce systematic bias, as lower thresholds would tend to classify more conditions as NOx-limited.

   The authors should clarify how this inconsistency is addressed. It would be helpful to add a separate section in which the FNRsec thresholds are first applied to GEOS-Chem–derived FNRsec, producing regime diagnostics analogous to Fig. 8, and then compare those results with the regime classification obtained using Pandora-derived FNR_all. This comparison would allow readers to better assess the implications of applying FNRsec thresholds to observational data.

   In addition, the thresholds are currently derived using data aggregated over the entire year. Given that ozone production efficiency is much lower during cold seasons, the

authors may consider focusing threshold derivation on the warmer months, when photochemical ozone production is most relevant, and focus the analysis on the warm season as well.

2. **Assumptions in separating secondary HCHO**
   The approach of turning off anthropogenic HCHO emissions to isolate secondary HCHO is reasonable. However, it implicitly assumes that the chemical processes controlling secondary HCHO formation are not significantly altered by the removal of primary HCHO sources. The authors should clarify whether and how this assumption holds, and discuss potential biases introduced by this approach. A sensitivity discussion with reference to studies that explicitly validate this methodology would strengthen confidence in the derived FNRsec.

3. **Interpretation of "external transport" removal**
   In Sect. 2.3, the manuscript states that external transport is "eliminated" by subtracting Run-3 or Run-4 from Run-2. This description is potentially misleading, as emission perturbation experiments do not strictly isolate transport processes. The authors should clarify a. which component of transport or background influence is being minimized by this subtraction, and b. how the resulting VOC and NOx emission reductions should be interpreted physically.

4. **Representativeness of in-situ surface measurements**
   Surface in-situ measurement is only available at Tokyo-TMU and only for limited periods (July and October 2022). Comparisons with surface $HCHO/NO_2$ derived from Pandora observations could help extent the surface regime diagnose.

5. **Policy relevance**
   The discussion of policy implications in the Conclusions is useful but remains somewhat general. The authors may consider adding a brief, concrete example to more directly connect the scientific findings with actionable air quality management strategies for Japan, particularly given the regional focus of the study.

**Technical corrections**

1. Consider adding a separate paragraph in the Introduction explicitly describing the motivation and importance of studying the vertical distribution of ozone sensitivity regimes within the troposphere.
2. Sections 3.1 and 3.2 could potentially be combined, as both primarily describe diurnal and seasonal patterns of HCHO and $NO_2$. In addition, it would be useful to explicitly

present the temporal evolution of FNR values themselves, since these are directly compared with thresholds to diagnose regimes.

3. To maintain consistency with the main results, the abstract should briefly mention the diurnal variation of ozone sensitivity regimes. Currently, only seasonality and vertical variations are mentioned.

4. Several instances of "nitrogen_dioxide" appear in the text; this should be consistently formatted as "nitrogen dioxide" or "$NO_2$."

5. Line 80: consider revising "extensive studies on $O_3$ formation are needed to efficiently mitigate human exposure" to "... needed to **more efficiently** mitigate human exposure," to reflect improvement relative to current mitigation effectiveness.